# STORI: A BENCHMARK AND TAXONOMY FOR STOCHASTIC ENVIRONMENTS

## ABSTRACT

Reinforcement learning (RL) techniques have achieved impressive performance on simulated benchmarks such as Atari100k, yet recent advances remain largely confined to simulation and show limited transfer to real-world domains. A central obstacle is environmental stochasticity, as real systems involve noisy observations, unpredictable dynamics, and non-stationary conditions that undermine the stability of current methods. Existing benchmarks rarely capture these uncertainties and favor simplified settings where algorithms can be tuned to succeed. The absence of a well-defined taxonomy of stochasticity further complicates evaluation, as robustness to one type of stochastic perturbation, such as sticky actions, does not guarantee robustness to other forms of uncertainty. To address this critical gap, we introduce STORI (STOchastic-ataRI), a benchmark that systematically incorporates diverse stochastic effects and enables rigorous evaluation of RL techniques under different forms of uncertainty. We propose a comprehensive five-type taxonomy of environmental stochasticity and demonstrate systematic vulnerabilities in state-of-the-art model-based RL algorithms through targeted evaluation of DreamerV3 and STORM. Our findings reveal that world models dramatically underestimate environmental variance, struggle with action corruption, and exhibit unreliable dynamics under partial observability. We release the code and benchmark publicly at `https://anonymous.4open.science/r/stori-353D`, providing a unified framework for developing more robust RL systems.

## 1 INTRODUCTION

Reinforcement learning (RL) techniques have achieved impressive performance on simulated benchmarks such as Atari100k, yet recent advances remain largely confined to simulation and show limited transfer to real-world domains. A central obstacle is environmental stochasticity, as real systems involve noisy observations, unpredictable dynamics, and non-stationary conditions that undermine the stability of current methods (Antonoglou et al., 2022; Paster et al., 2022). This challenge is especially acute for model-based RL, which must build world models to capture environment dynamics, a task that becomes significantly more complex when the environment exhibits multiple forms of uncertainty.

However, we lack a comprehensive stochastic environment benchmark that enables systematic development of RL methods robust to environmental stochasticity. Most widely used benchmarks, such as Atari games in the Arcade Learning Environment (ALE) (Bellemare et al., 2013), are deterministic or nearly deterministic (Paster et al., 2022). Although several approaches have introduced limited stochasticity, including sticky actions (Machado et al., 2018), no-ops (Mnih et al., 2015), human starts (Nair et al., 2015), and random frame skips (Brockman et al., 2016)—these modifications remain narrow in scope. To develop truly robust RL agents, we need benchmarks that systematically incorporate diverse forms of environmental uncertainty with granular control over both types and intensities of stochastic effects.

In this paper, we introduce STORI (STOchastic-ataRI), a benchmark that systematically incorporates diverse stochastic effects and enables rigorous evaluation of RL techniques under different forms of uncertainty. Alongside, we propose an updated taxonomy of stochasticity in RL environments, providing a unified framework for analyzing and comparing approaches. We leverage STORI to systematically investigate the reliability of world models under diverse forms of stochasticity and

perform targeted evaluation probes to examine how learned dynamics respond to different stochastic challenges.

Our key contributions include:

- **A comprehensive stochasticity taxonomy** with five distinct types: action-dependent noise, action-independent randomness, concept drift, representation learning challenges, and missing state information
- **STORI benchmark implementation** that systematically incorporates these stochasticity types into four Atari environments with granular parameter control
- **Systematic evaluation** of state-of-the-art model-based RL algorithms (DreamerV3 and STORM) revealing fundamental vulnerabilities to environmental uncertainty
- **Targeted failure mode analysis** demonstrating that world models systematically underestimate variance, struggle with action corruption, and show unreliable dynamics under partial observability
- **Open-source framework** enabling researchers to develop and evaluate stochasticity-aware RL algorithms. We release the code and benchmark publicly at `https://anonymous.4open.science/r/stori-353D`

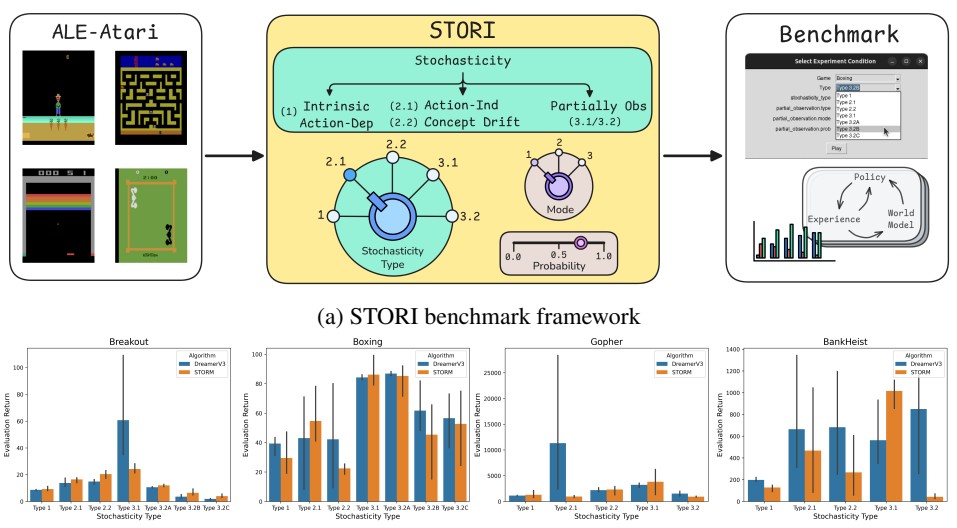

(a) STORI benchmark framework

(b) Performance across stochasticity types

Figure 1: STORI benchmark and results. (a) Framework for systematic stochasticity evaluation. (b) DreamerV3 and STORM performance degradation under uncertainty (Types: 1=action corruption, 2.1=random events, 2.2=concept drift, 3.1=default, 3.2=missing information).

## 2 RELATED WORKS

**Stochastic Environment Benchmarks** Recent efforts have addressed limitations of deterministic RL benchmarks by incorporating stochastic perturbations. Robust-Gymnasium (Gu et al., 2025) provides a modular framework for robust evaluation across sixty robotics and control tasks, introducing observation-disruptors, action-disruptors, and environment-disruptors for systematic robustness assessment. Similarly, Zouitine et al. (2024) introduced a benchmark extending Gymnasium-MuJoCo with six tasks capturing environmental shifts. STORI shares similar motivations but offers complementary contributions. First, it uses Atari games as a canonical testbed for discrete, high-dimensional decision-making. Second, STORI explicitly includes temporal non-stationarities such as concept drift, a critical yet underexplored aspect of real-world uncertainty. Beyond comprehensive benchmarks, several works target specific perturbation types. Zhang et al. (2020) examined adversarial state perturbations, Han et al. (2024) studied multi-agent adversarial attacks, and Park et al. (2025) analyzed offline goal-conditioned RL under perturbations. These reveal vulnerabilities

to specific noise sources but lack unified robustness suites. Sandbox frameworks like MiniHack (Samvelyan et al., 2021) allow custom stochastic environments.

**Stochasticity Taxonomy** Early RL literature formalized uncertainty through Partially Observable Markov Decision Processes (POMDPs) (Sutton & Barto, 2018), addressing partial observability and stochastic transitions. Recent works like Vamplew et al. (2022) propose broader stochastic MDP classifications for policy evaluation and learning. Beyond formal stochasticity, Liu et al. (2023) introduces environment heterogeneity concerning spatial layout and dynamics variations. Lu et al. (2018) addresses temporal non-stationarity or concept drift, where target concepts change due to shifting hidden contexts. STORI's taxonomy integrates existing perspectives into a unified, practical framework explicitly designed for benchmarking, allowing systematic instantiation of stochasticity classes as configurable perturbations.

**World Model Benchmarks** Recent world modeling advances include DreamerV3 (Hafner et al., 2024), IRIS (Micheli et al., 2023), STORM (Zhang et al., 2023), TransDreamer (Chen et al., 2024), TWM (Robine et al., 2023), and DIAMOND (Alonso et al., 2024). These have been evaluated using Atari 100k (Ye et al., 2021), BSuite (Osband et al., 2020), Crafter (Hafner, 2021), and DMLab (Beattie et al., 2016). STORI introduces a stochasticity-driven framework to analyze world model performance under environmental uncertainties.

## 3 ENVIRONMENT STOCHASTICITY AND OUR MOTIVATION

### 3.1 ENVIRONMENT STOCHASTICITY

RL environments can be categorized by their predictability. Deterministic environments have fully predictable outcomes, while stochastic environments introduce uncertainty requiring agents to handle outcome variability. Following Kumar & Varaiya (1986), stochasticity includes: **Stationary Stochastic (Objective)** with consistent statistical distributions (coin tosses); **Stationary Stochastic (Subjective)** based on personal beliefs (expert forecasts); **Non-Stationary Stochastic** with time-varying dynamics (traffic patterns); and **Illusory or Complex Uncertainty** where probabilities are unreliable (nuclear accidents).

### 3.2 MATHEMATICAL TAXONOMY OF STOCHASTICITY

We formalize five key types using transition function $P(s'|s, a)$ where $s$ is the current state, $a$ is the action, and $s'$ is the next state.

#### 3.2.1 TYPE 1: INTRINSIC ACTION-DEPENDENT STOCHASTICITY

Unreliable action channels corrupt intended action $a$ into executed action $\tilde{a}$.

$$P_{AD}(s'|s, a) = \sum_{\tilde{a} \in \mathcal{A}} C(\tilde{a}|a) P(s'|s, \tilde{a}) \tag{1}$$

For sticky actions: $C(\tilde{a}|a) = (1 - \alpha)\mathbb{I}\{\tilde{a} = a\} + \alpha \cdot u(\tilde{a})$

#### 3.2.2 TYPE 2.1: INTRINSIC ACTION-INDEPENDENT STOCHASTICITY (RANDOM)

Exogenous random events independent of agent actions, with random variable $\xi \sim F(\xi)$:

$$P_{AI}(s'|s, a) = \int P(s'|s, a, \xi) dF(\xi) \tag{2}$$

#### 3.2.3 TYPE 2.2: INTRINSIC ACTION-INDEPENDENT STOCHASTICITY (CONCEPT DRIFT)

Time-dependent dynamics $P_t(s'|s, a)$ with drift magnitude:

$$\text{Drift}(t, t + \Delta t) = \mathcal{D}(P_t(\cdot|s, a)||P_{t+\Delta t}(\cdot|s, a)) \tag{3}$$

#### 3.2.4 TYPE 3.1: AGENT-INDUCED STOCHASTICITY (REPRESENTATION LEARNING)

Rich observations requiring representation learning where $I(S; O) \approx H(S)$. Belief states update via:

$$b_{t+1}(s') \propto O(o_{t+1}|s') \sum_{s \in \mathcal{S}} P(s'|s, a_t) b_t(s) \tag{4}$$

State aliasing is resolvable through better feature extraction. *Example:* Standard Atari RGB frames contain all game information but require learning to extract from pixels.

### 3.2.5 TYPE 3.2: AGENT-INDUCED STOCHASTICITY (MISSING STATE VARIABLES)

Critical state variables are completely omitted where $I(S; O) \ll H(S)$. Creates fundamental state aliasing $O(o|s_i) = O(o|s_j) = 1$ for $s_i \neq s_j$ that persists regardless of representational capacity. Same belief update as Type 3.1 but fundamentally limited.

Requires history tracking: $h_t = \{o_1, a_1, o_2, a_2, \ldots, o_t\}$

*Examples:* Breakout with invisible ball regions; Boxing with hidden opponents; partial screen occlusion.

### 3.3 CHALLENGES FOR WORLD MODEL LEARNING AND MODEL BASED RL IN STOCHASTIC ENVIRONMENTS

In this section, we analyze potential challenges for MBRL in different types of stochastic environments. A world model, denoted $\tilde{P}_\theta$, aims to learn the true transition dynamics from data. Each form of stochasticity introduces a distinct challenge that can cause a mismatch between $\tilde{P}_\theta$ and the true dynamics.

**Challenge from Type 1 Stochasticity** The world model must learn not only the environment's response to actions, $P(s'|s, \tilde{a})$, but also the action channel itself, $C(\tilde{a}|s, a)$. If the model fails to account for the action channel, its predictions will be systematically biased. The prediction error is the divergence between the model's direct prediction and the true, action-corrupted dynamics: Error $= \mathcal{D}(P_{AD}(s'|s, a) || \tilde{P}_\theta(s'|s, a))$, where $P_{AD}$ represents the true dynamics and $\tilde{P}_\theta$ represents the model prediction. The model's ability to control the environment is limited by the **action channel capacity**, which can be measured by the mutual information $I(A; \tilde{A}|S)$. A low-capacity channel is fundamentally difficult to model and exploit.

**Challenge from Type 2.1 Stochasticity** This introduces irreducible **aleatoric uncertainty** into the environment. A deterministic world model will fail completely. A probabilistic world model must accurately capture the variance of the outcomes. The world model must learn a distribution over next states. The core challenge is to match the variance of this distribution to the true environmental variance, which is inherent and cannot be reduced with more data. The prediction error is tied to the model's ability to capture this spread: Aleatoric Error $= |\text{Var}_{s' \sim P_{AI}}[s'] - \text{Var}_{s' \sim \tilde{P}_\theta}[s']|$. The world model must avoid being overconfident in its predictions and instead represent the full range of possible outcomes.

**Challenge from Type 2.2 Stochasticity** Concept drift causes the world model's learned parameters $\theta$ to become outdated. A model trained on data from time $t$ will be inaccurate at time $t + \Delta t$. The prediction error grows over time as the environment drifts away from the data the model was trained on. The accumulated error is a function of the drift magnitude: Prediction Error$(t + \Delta t) \propto \mathcal{D}(P_t(\cdot|s, a) || \tilde{P}_\theta(\cdot|s, a))$, where $\tilde{P}_\theta$ was trained on data from distributions around time $t$. This forces the model to either continuously adapt its parameters or have a mechanism to detect and react to the drift.

**Challenge from Type 3.1 & 3.2 Stochasticity** The world model cannot operate on the true state $s$ and must instead learn a latent state representation $z_t$ from a history of observations $o_{1:t}$. The primary challenge is **state aliasing**. The uncertainty a world model faces is not just the environment's true stochasticity, but also the aliasing-induced variance. The total variance in outcomes given an observation $o$ is decomposed as: $\text{Var}[s'|o, a] = \mathbb{E}_{s \sim b(s|o)}[\text{Var}[s'|s, a]] + \text{Var}_{s \sim b(s|o)}[\mathbb{E}[s'|s, a]]$, where the first term represents true aleatoric uncertainty and the second term represents aliasing-induced uncertainty. The world model's latent dynamics, $\tilde{P}_\theta(z'|z, a)$, must implicitly handle the second term, which is purely an artifact of perception. In Type 3.2 environments, where entire state variables are missing, this aliasing uncertainty can become overwhelmingly large, making it nearly impossible to form an accurate belief state and rendering long-term prediction unreliable.

## 4 STORI - A BENCHMARK OF STOCHASTIC ENVIRONMENTS FOR RL

In this section, we describe in details the benchmark environments we built for different types of stochasticity based on Atari-Arcade learning environment. Atari games such as Breakout, Boxing, Gopher and BankHeist were modified to allow fine-grain control of these stochasticity. The taxonomy for stochasticity in STORI is an extension of the summary of classification of stochasticity according to Antonoglou et al. (2022).Table 1 presents the taxonomy of stochasticity, listing

each type, its corresponding subtype, and the associated ID. Each type is explained in the following sections.

Table 1: Environment stochasticity taxonomy

| ID | Type | Sub Type |
|----|------|----------|
| 0 | Deterministic | NA |
| 1 | Action Dependent | NA |
| 2.1 | Action Independent | Random |
| 2.2 | Action Independent | Concept Drift |
| 3.1 | Partially Observed | Representation |
| 3.2 | Partially Observed | Missing State |

### ATARI - ARCADE LEARNING ENVIRONMENT

The Arcade Learning Environment (ALE) provides a foundational framework for applying RL to Atari 2600 games (Bellemare et al., 2013). Built on the Stella emulator and integrated with Gymnasium (Brockman et al., 2016), ALE supports over a hundred games with extensive configurability including observation types (RGB, grayscale, RAM), action spaces, and stochasticity parameters like sticky actions (Machado et al., 2018). The Atari 100K benchmark (Ye et al., 2021) evaluates sample efficiency by assessing agents after only 100,000 environment steps (approximately two hours of gameplay).

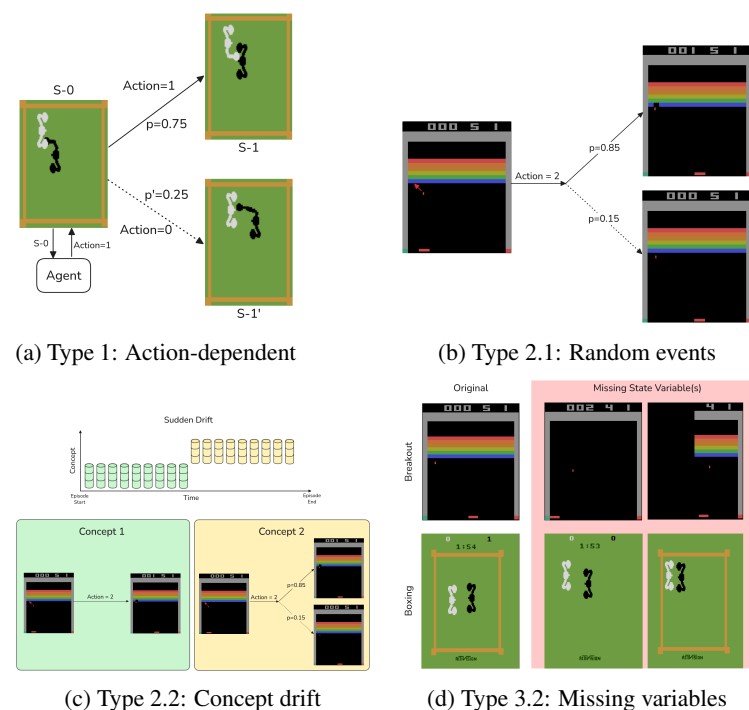

(a) Type 1: Action-dependent

(b) Type 2.1: Random events

(c) Type 2.2: Concept drift

(d) Type 3.2: Missing variables

Figure 2: Stochasticity types in STORI benchmark.

### TYPE 0: DETERMINISTIC ENVIRONMENT

Deterministic environments are those in which the next state is fully determined by the current state and the action taken. The state is completely observable and there is no randomness in the state transitions or rewards, meaning that the outcome of any action is predictable.

In the case of Atari, we consider the ground-truth labels of various state variables obtained directly from the RAM for each observation, following the approach of Anand et al. (2020). No additional

stochasticity parameters are introduced, meaning that the environment is fully deterministic and corresponds to Type 0 in our taxonomy. Example for Breakout can be seen in figure 5.

### TYPE 1: INTRINSIC ACTION DEPENDENT STOCHASTIC ENVIRONMENT

In environments with action-dependent intrinsic stochasticity, the environment may replace the agent's chosen action with different action, by default, with a random one. For instance, in $sticky\_action$ (Machado et al., 2018) scenarios, the environment can repeat the previous action with some probability. This results in varied outcomes even from the same state, with the stochastic effects limited to the state variables that can be influenced by the agent's actions. An example of action-dependent intrinsic stochasticity with Atari Boxing can be seen in the 2a.

### TYPE 2.1: INTRINSIC ACTION INDEPENDENT STOCHASTIC ENVIRONMENT - RANDOM

In action-independent random stochastic environments, randomness arises independently of the agent's choices and affects state variables outside the agent's direct control. This stochasticity, often due to external factors or inherent environmental noise, means that even with complete knowledge of the environment and carefully chosen actions, the next state cannot be predicted with certainty.

The figure 2b illustrates an example of how this type of stochasticity can be modeled in Atari Breakout, where the paddle is moved to the right while the ball is on a trajectory to hit a block. In this case, there is a 0.15 probability that the ball bounces back without destroying the block or yielding any reward. Notably, this stochastic behavior is independent of the action of moving the paddle to the right.

### TYPE 2.2: INTRINSIC ACTION INDEPENDENT STOCHASTIC ENVIRONMENT - CONCEPT DRIFT

Environments with intrinsic action-independent concept drift can change over time independently of the agent's actions, a phenomenon known as concept drift (Lu et al., 2018), which can generally be categorized into three types according to how the drift unfolds over time. In *sudden drift*, the environment undergoes abrupt changes, forcing the agent to quickly adapt to new dynamics. In *gradual or incremental drift*, the transition to new dynamics occurs slowly over time, requiring the agent to continuously adjust its policy. Finally, in *recurring drift*, previously observed dynamics reappear in a cyclical or context-dependent manner, making long-term adaptation more challenging. Learning in such environments demands flexibility and the ability to detect and respond to changes.

In the case of Atari, most games have intrinsic incremental drift. As the agent levels up in the game, the difficulty of the game also increases. With a more fine-grain control over concept drift, other types of drift can also be achieved in Atari games as shown in figure 2c.

### TYPE 3.1: PARTIALLY OBSERVED ENVIRONMENT (REPRESENTATION LEARNING)

In partially observed environments, the agent does not have access to the full state information. When the state variables are represented differently from the true underlying environment, the agent must infer hidden information or learn an appropriate representation. This increases the complexity of decision making, since the agent must rely on approximate observations.

A typical example is the **Default Atari setting**, where the agent perceives only the screen image produced by the emulator after each action. These images are designed to approximate the true state, but they do not capture it fully. To enrich the observation, many implementations use a 4-frame skip with aggregation, which allows the agent to infer additional information, such as motion or rate of change over time, that is not apparent from a single frame.

### TYPE 3.2: PARTIALLY OBSERVED ENVIRONMENT (MISSING STATE VARIABLE(S))

An important subclass of partially observed environments arises when information about certain state variables is missing, leaving the agent unable to observe critical aspects of the environment. This lack of information demands strategies that can manage uncertainty and make robust decisions despite gaps in perception. Such environments are common in real-world scenarios where sensors are limited, noisy, or unreliable.

Figure 2d illustrates type 3.2 environments using Atari Breakout and Boxing. In Breakout, examples include invisible blocks or a partially hidden screen, while in Boxing, examples include a hidden boxing ring or concealed clock and score information.

## 5 EXPERIMENTS AND RESULTS

We evaluated DreamerV3 (Hafner et al., 2024) and STORM (Zhang et al., 2023) using STORI. DreamerV3 features a learned world model with actor-critic architecture, achieving robustness through fixed hyperparameters, normalization, and effective scaling. STORM employs a transformer backbone with stochastic variational modeling for strong sequence modeling and robustness. We focus on MBRL methods as model-free approaches require substantially more computational resources for meaningful results.

We selected four Atari 100K environments: *Breakout*, *Boxing*, *Gopher* (Agent-Optimal (Lim et al., 2025)), and *BankHeist* (Human-Optimal (Lim et al., 2025)). Breakout, Boxing, and Gopher are classified as Agent-Optimal tasks, where agents achieve strong performance under default conditions, facilitating the study of performance degradation under stochasticity. BankHeist, a Human-Optimal task, provides a preliminary understanding of how added stochasticity interacts with environments where agents perform substantially worse than humans. The set also offers diversity in action-space sizes: Breakout (4 actions), Gopher (8 actions), and Boxing/BankHeist (18 actions each), ensuring that robustness observations are not biased by a particular action-space complexity.

For each environment, stochasticity types were assigned mostly at random, except for Type 2.2 (concept drift), where Concept 2 was intentionally selected from other stochasticity types to enable controlled comparison between "pure" stochasticity and its drifting variant. Multiple parameter settings for each stochasticity type and game were provided, allowing users of the benchmark to select perturbation regimes relevant to their research focus. Implementation details are in Appendix A.1, and experiment-specific stochasticity settings are provided in Appendix B.1. Each algorithm was trained for 100K steps across baseline and modified environments for three seeds and evaluated on 100 episodes, with mean return reported. Note that Type 3.1 is the default ALE environment setting.

### 5.1 PERFORMANCE OF DREAMERV3 AND STORM IN DIFFERENT STOCHASTIC ENVIRONMENTS

#### 5.1.1 BREAKOUT

Stochasticity introduction caused marked performance decline versus default Type 3.1 environment (Figure 1b), aligning with theoretical predictions. DreamerV3 initially outperformed STORM (60.71±41.89 vs 24.17±3.55) but STORM showed greater robustness across stochasticity types.

Table 2: Variance underestimation by world models.

| Model | Type | Diff. |
|-------|------|-------|
| DreamerV3 | 3.1 | 1.25 |
| DreamerV3 | 2.1 | 300.34 |
| STORM | 3.1 | 1.32 |
| STORM | 2.1 | 325.21 |

Breakout's small action space (4 actions) creates high sensitivity to perturbations as incorrect LEFT/RIGHT actions immediately cause failure. Unlike Boxing, Breakout offers no recovery margin, amplifying uncertainty's long-term impact.

Type 2.1 environments caused severe struggles, with performance dropping to 15% of baseline, confirming that irreducible aleatoric uncertainty challenges deterministic world models. Type 2.2 concept drift (default→Type 3.2A after 300 steps) showed better performance than standalone Type 3.2A, suggesting adaptive mechanisms can leverage temporal structure.

#### 5.1.2 BOXING

Boxing showed less pronounced performance decline. STORM initially led (86.18±11.29 vs 84.22±1.68) but DreamerV3 outperformed across several stochasticity types. Boxing's resilience stems from: (1) larger action space (18 actions) providing redundancy with functionally similar actions, and (2) recovery mechanisms through retreating/repositioning.

Type 3.2A (hidden score/clock) counterintuitively improved DreamerV3 performance (86.90±1.33 vs 84.22±1.68), suggesting non-essential information removal simplifies representation learning.

For Type 3.2B (75% right-half occlusion), agents showed adaptive behavior as they confined opponents to visible areas, transforming partial observability into strategic constraints.

Type 2.2 concept drift performed worse than standalone Type 3.2C, except DreamerV3's third seed learned to maximize early-episode scores before opponent invisibility, demonstrating strategic adaptation to predictable timing.

### 5.1.3 GOPHER AND BANKHEIST

Gopher showed high variability with Type 2.1 producing anomalous DreamerV3 performance (11,333.53±14,761.12) due to beneficial reward cancellation dynamics—suggesting implementation-specific edge case exploitation.

BankHeist exhibited divergent Type 3.2 outcomes: DreamerV3 achieved 849.80±514.15 while STORM fell to 43.10±22.85. DreamerV3 consistently adopted an unconventional policy, remaining near city gates and triggering inter-city transitions to loot nearby banks while avoiding stochastic modifications—effectively exploiting structural features to reduce tasks to near-deterministic sub-problems rather than demonstrating genuine robustness. STORM explored broadly within cities, exposing itself to full stochastic impact. This highlights that high returns may reflect reward-maximizing shortcuts exploiting environment dynamics rather than genuine uncertainty resilience.

## 5.2 ANALYSIS OF ERROR TYPES AND WORLD MODEL FAILURES

To understand the specific failure modes of model-based RL under different stochasticity types, we conducted targeted analyses for some error categories defined in Section 3.3.

### 5.2.1 ERRORS CAUSED BY TYPE 2.1 STOCHASTICITY: ALEATORIC UNCERTAINTY

We ran a controlled probe of a single repeated action (action 3) for 1000 steps in both Type 3.1 (default setting,partially observed) and Type 2.1 (action-independent stochasticity) BankHeist environments, collecting states from the environment and predictions from the world models from DreamerV3 and STORM (Table 2). The resulting variance differences include:

**Environment variance difference:**

$$\text{Var}_{\text{env}}(\text{Type 2.1}) - \text{Var}_{\text{env}}(\text{Type 3.1})$$

$\rightarrow$ DreamerV3: 299.097, STORM: 323.904. This confirms that Type 2.1 environments exhibit significantly higher true variance due to stochasticity.

**Model variance difference:**

$$\text{Var}_{\text{model}}(\text{Type 2.1}) - \text{Var}_{\text{model}}(\text{Type 3.1})$$

$\rightarrow$ DreamerV3: 0.00465, STORM: 0.01216. Both models predict nearly identical variance between environments despite the true variance increasing substantially.

Both DreamerV3 and STORM significantly underestimate the increased stochasticity present in Type 2.1 environments in BankHeist. While environment variance increases by approximately 300, the models' predicted variances remain nearly constant. This mismatch highlights a lack of variance calibration under action-independent stochastic conditions, revealing a limitation in the world models' ability to capture environment uncertainty accurately.

### 5.2.2 ERRORS CAUSED BY TYPE 2.2 STOCHASTICITY: CONCEPT DRIFT ANALYSIS

For concept drift stochasticity (*BankHeist* Type 2.2), we measured the degradation ratio of model performance before and after the drift point.

The degradation ratio is defined as:

$$\text{Degradation ratio (dyn\_loss)} = \frac{\text{dyn\_loss}_{\text{Concept 2}}}{\text{dyn\_loss}_{\text{Concept 1}}}$$

Both DreamerV3 and STORM exhibit substantial increases in dynamics loss after the concept change, indicating that world model accuracy deteriorates significantly when the environment transitions to Concept 2.

**STORM – BankHeist (Type 2.2):**

- Concept 1: dyn_loss $= 4.51 \pm 2.42$
- Concept 2: dyn_loss $= 22.71 \pm 4.20$
- Degradation ratio = 5.04

**DreamerV3 – BankHeist (Type 2.2):**

- Concept 1: dyn_loss $= 5.43 \pm 15.86$
- Concept 2: dyn_loss $= 37.67 \pm 36.37$
- Degradation ratio = 6.94

These results highlight that both model-based agents experience a sharp decline in world model performance following the concept drift, emphasizing the challenge of adapting to non-stationary dynamics.

### 5.2.3 Errors caused by type 3 stochasticity: State Aliasing Effects

To investigate how well world models handle missing information, we designed a controlled experiment using BankHeist Type 3.2, where city blocks are randomly hidden in 75% of observations. The key question: does a model's prediction accuracy depend on whether it can initially see the environment clearly?

**Experimental design:** We created six test scenarios and compared two starting conditions for each:

- **Clear-start**: Model begins with city blocks visible, takes an action, observes the result
- **Obscured-start**: Model begins with city blocks hidden, takes the same action, sees the same result

We then measured how "surprised" each model was by computing the difference in prediction error: $\Delta$NLL = NLL(obscured-start) $-$ NLL(clear-start) as in figure 3.

**Key findings:** As in table 3, DreamerV3 shows positive $\Delta$NLL values (1.15), meaning it makes significantly worse predictions when starting from obscured observations. In contrast, STORM shows negative $\Delta$NLL values ($-3.32$), indicating it actually performs slightly better when starting from limited information.

This reveals that DreamerV3's world model relies heavily on having complete initial observations to make accurate predictions. When city blocks are initially hidden, DreamerV3 struggles to maintain accurate beliefs about the environment state, requiring larger "corrections" to its internal model after seeing the action's outcome.

**Critical insight:** High task performance does not guarantee robust world model dynamics. Despite achieving strong returns in partially observable environments, DreamerV3's world model is more brittle when dealing with missing information compared to STORM.

## 6 Limitations and Future Work

Table 3: Model prediction errors under partial observability.

| Model | $\Delta$NLL | $\Delta$KL |
|---|---|---|
| DreamerV3 | 1.15±2.46 | -0.18±2.83 |
| STORM | -3.32±2.86 | 0.18±0.62 |

**Limited coverage of model-free baselines.** Our benchmark currently focuses on DreamerV3 and STORM, two state-of-the-art model-based algorithms. This choice was driven by practical constraints of the evaluation setting: we study robustness in a 100K-interaction regime across multiple stochasticity types, magnitudes, and four Atari games, resulting in substantial computational cost. Many widely used model-free agents

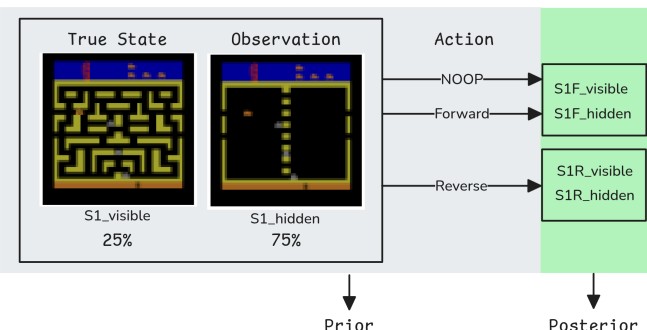

Figure 3: Partial observability probe showing prediction errors when models start with clear vs. obscured observations.

(e.g., PPO, Rainbow, IQN) are not competitive under 100K steps, as they typically require millions of frames for reasonable performance. Including them would therefore provide limited insight into robustness. Although fast-learning model-free methods such as BBF ("Bigger, Better, Faster") exist, the public JAX implementation did not integrate cleanly with our PyTorch-based evaluation pipeline, leading to compatibility overheads. We acknowledge the value of including sample-efficient model-free algorithms and are actively extending our framework to benchmark such agents in future versions.

**Cross-type comparison constraints.** Our results focus on within-type rather than cross-type robustness comparisons. This reflects the fact that cross-type comparisons are meaningful only when reward semantics remain consistent across stochasticity types of a given environment. For `Boxing` and `BankHeist`, all stochasticity types preserve reward consistency and are directly comparable. For `Breakout` and `Gopher`, all types except Type 2.1 satisfy this criterion. In Type 2.1 for these games, perturbations (e.g., stochastically canceled hits) fundamentally alter the achievable returns and distort the optimal reward distribution. Such shifts make the environments structurally incomparable to other types; reporting cross-type results in these settings would be misleading.

**Future directions.** A promising extension of this work is to evaluate agents under *compositional* uncertainty, where multiple stochasticity types are activated simultaneously. This would more faithfully represent real-world uncertainty, where observation noise, transition randomness, and action corruption may interact in nontrivial ways. Another direction is to benchmark uncertainty-aware agents, including Bayesian world models, ensemble-based approaches, adversarially trained policies, and domain-randomization methods. Studying such techniques within our stochasticity taxonomy may yield deeper insights into how agents can model, anticipate, and adapt to diverse forms of uncertainty.

## 7 CONCLUSIONS

We introduced STORI, a systematic benchmark with a five-type taxonomy for evaluating RL algorithms under environmental stochasticity: action-dependent noise, action-independent randomness, concept drift, representation learning challenges, and missing state information.

Evaluation of DreamerV3 and STORM revealed systematic vulnerabilities in model-based approaches. Both algorithms struggle with action corruption, underestimate environmental variance by 300×, degrade 5-7× after concept drift, and show inconsistent reliability under partial observability. Strong task performance does not guarantee robust world model dynamics. STORI provides a foundation for building more robust RL systems capable of handling real-world uncertainty.

## 8 REPRODUCIBILITY STATEMENT

We are committed to ensuring the reproducibility of our findings. All data, code, and implementation details necessary to replicate our experiments will be made available to the research community. Careful documentation accompanies the released resources to facilitate independent verification and reuse. The authors affirm that the results reported in this paper can be fully reproduced using the provided materials.

## 9 ETHICS STATEMENT

This work was conducted in accordance with established ethical standards for scientific research. All methods, analyses, and interpretations were carried out with a commitment to transparency, integrity, and responsible reporting. The authors confirm that no part of this research involved practices that could compromise fairness, safety, or the ethical treatment of data, participants, or systems.

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

# A APPENDIX

## A.1 STORI IMPLEMENTATION

The STORI framework is built around a sophisticated wrapper-based architecture that introduces various types of uncertainty and partial observability into deterministic Atari environments with a granular control over the modifications.

### A.1.1 CORE ARCHITECTURE AND WRAPPER SYSTEM

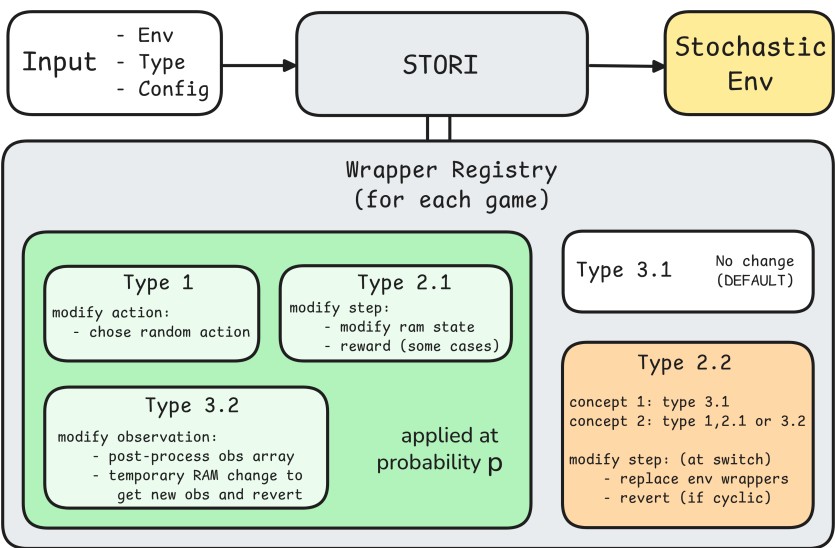

Figure 4: STORI Implementation Overview

The implementation uses a hierarchical wrapper system built on top of the Atari Learning Environment (ALE). The main 'StochasticEnv' class serves as the entry point, which applies different types of wrappers from 'wrapper_registry' of specified environment. The system supports five distinct types, each introducing different forms of stochasticity. The system is highly configurable through a dictionary-based configuration system. Users can specify probabilities for different stochasticity effects, choose between different modes of operation, and configure temporal parameters for concept drift. The wrapper registry system allows for easy extension and customization of stochasticity types for new games or research requirements.

### A.1.2 STOCHASTICITY WRAPPERS

- Type 0: This type returns the RAM state of the game (a 1-D numpy array) with state labels as the observation. This implementation is an extension of Atari Annotated RAM Interface (Anand et al., 2020).

- Type 1: The 'ActionDependentStochasticityWrapper' randomly replaces the agent's intended action with a random action from the action space with a specified probability.

- Type 2.1: The 'ActionIndependentRandomStochasticityWrapper' implements environment specific random events that occur independently of the agent's actions. These effects are applied probabilistically and create unpredictable environmental changes to which the agent must adapt. Read more about game-specific modifications in section D.

- Type 2.2: This introduces temporal concept drift where the environment dynamics change over time. The 'ActionIndependentConceptDriftWrapper' supports both sudden and cyclic modes between 2 concepts. The concept 1 is the default environment (type 3.1) and concept 2 can be any other environment stochasticity types out of 1, 2.1 and 3.2. In sudden mode,

the environment switches to concept 2 after a fixed number of steps. In cyclic mode, it alternates between the concept 1 and 2 every specified number of steps, creating a challenging environment where the agent must continuously adapt to changing dynamics.

- Types 3.1: This stochasticity type returns the default ALE environment without any modifications.

- Types 3.2: The 'PartialObservationWrapper' introduces partial observability by modifying the agent's observations. The system supports multiple observation modification techniques including cropping (removing portions of the screen), blackout (hiding specific regions), and RAM manipulation (temporarily modifying the game's internal state to get modified observation).

In STORI, stochasticity types 1, 2.1, 2.2, and 3.2 are implemented as extensions of Type 3.1 environments. This is because screen-based observations serve as the default, well-studied ALE inputs for various reinforcement learning algorithms, providing a consistent foundation for comparing different types of stochasticity while also allowing for interpretable analysis of agent actions and behaviors.

### A.1.3 ALGORITHMS ADDITIONAL DETAILS

- DreamerV3: The source implementation and default parameters for Atari100K config used from this code repository (MIT license): `https://github.com/NM512/dreamerv3-torch`

- STORM: The source implementation and default parameters (except eval num_episode was set to 100) used from this code repository: `https://github.com/weipu-zhang/STORM`

## B ADDITIONAL BENCHMARK DETAILS

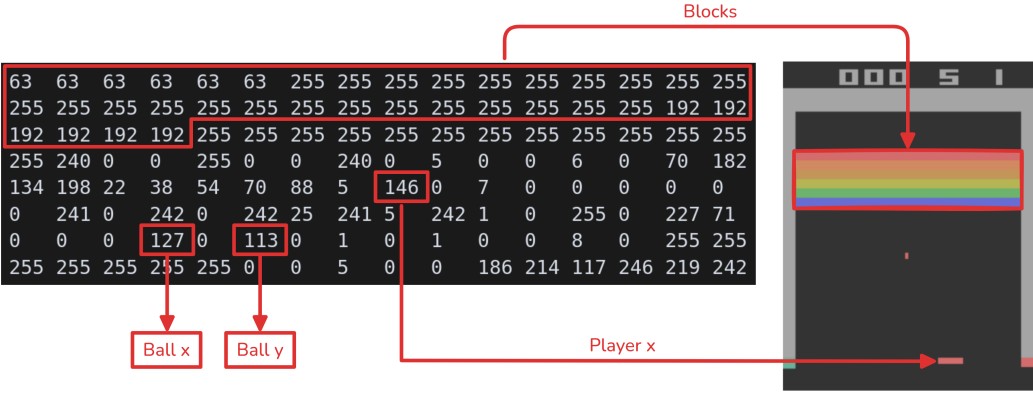

Figure 5: The figure shows the RAM state of Atari Breakout on the left and corresponding observation image from the emulator on the right, along with annotations for various state variables like ball position, blocks state etc.

## B.1 Experiment Stochasticity Modes

### B.1.1 Stochasticity Modes Used in Breakout Experiments

- Type 1: Random action executed from action space instead of predicted action with a probability of $0.3$.
- Type 2.1: If a block is hit, there is probability of $0.25$ that the hit is not considered and the block is not destroyed thereby returning 0 reward and the ball bounces back.
- Type 2.2: Episode starts with default setting (Type 3.1) and after 300 steps into the episode, the dynamics suddenly change to *Type 3.2A*.
- Type 3.1: Default Atari Breakout.
- Type 3.2A: The ball is only visible is a specific window between the blocks and the paddle and permanently hidden ($p = 1.0$) in rest of the space between them.
- Type 3.2B: Randomly hide left vertical half of the screen 75% ($p = 0.75$) of the episode.
- Type 3.2C: Only a random circular area of the screen is visible every frame ($p = 1.0$) similar to what someone will see when walking in a dark room with a torch.

### B.1.2 Stochasticity Modes Used in Boxing Experiments

- Type 1: Random action executed from action space instead of predicted action with a probability of $0.3$.
- Type 2.1: Swaps the color of the enemy and player (character and score) with probability of $0.001$ which results in 6-7 persistent swaps per episode (2 mins boxing round).
- Type 2.2: Episode starts with default setting (Type 3.1) and after 300 steps into the episode, the dynamics suddenly change to *Type 3.2C*.
- Type 3.1: Default Atari Boxing.
- Type 3.2A: Permanently hide ($p = 1.0$) scores and game clock.
- Type 3.2B: Randomly hide right vertical half of the screen 75% ($p = 0.75$) of the episode.
- Type 3.2C: Randomly hide enemy character 70% ($p = 0.7$) of the episode.

### B.1.3 Stochasticity Modes Used in Gopher Experiments

- Type 1: Random action executed from action space instead of predicted action with a probability of $0.3$.
- Type 2.1: Hole doesn't fill underground below the farmer and the reward is reverted to 0 whenever farmer digs, with probability of $0.3$.
- Type 2.2: At the beginning of each episode, the environment is set to the default mode (Type 3.1). Every 600 steps, the dynamics transition *cyclically* between Type 3.2 and the default.
- Type 3.1: Default Atari Gopher.
- Type 3.2: Permanently hide ($p = 1.0$) underground gopher movement and holes and only hole openings are visible on surface (if any).

### B.1.4 Stochasticity Modes Used in BankHeist Experiments

- Type 1: With probability $0.3$, a random action is executed from a restricted subset of the action space (0–9) instead of the predicted action. The restriction reduces the frequency of fire-based actions during random sampling, preventing the agent from instantly dying by triggering a bomb it drops on itself.
- Type 2.1: With probability $0.001$, the robber is unexpectedly teleported to a different city.
- Type 2.2: At the beginning of each episode, the environment is set to the default mode (Type 3.1). Every 600 steps, the dynamics transition *cyclically* between Type 3.2 and the default.
- Type 3.1: Default Atari BankHeist.
- Type 3.2: Randomly hide city blocks 75% ($p = 0.75$) of the frames.

## B.2 LEARNING CURVES FOR DIFFERENT STOCHASTICITY TYPES

Figures 6, 7, 8, and 9 illustrate the learning curves on Breakout, Boxing Gopher and BankHeist respectively, depicting the average evaluation return as a function of training steps up to 100K, for DreamerV3 and STORM.

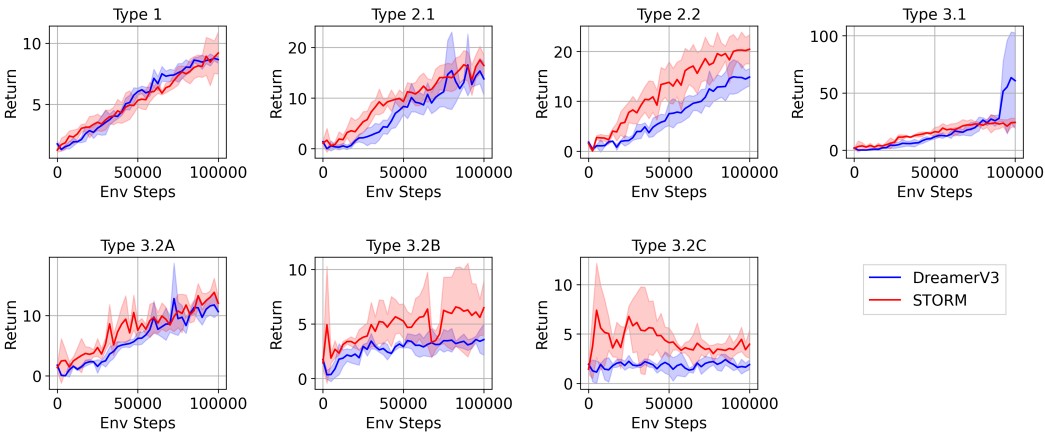

Figure 6: Breakout - learning curves

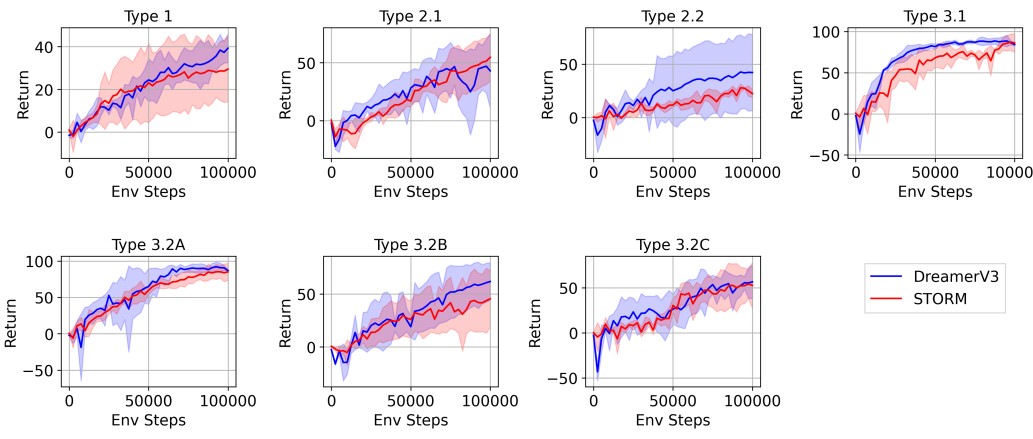

Figure 7: Boxing - learning curves

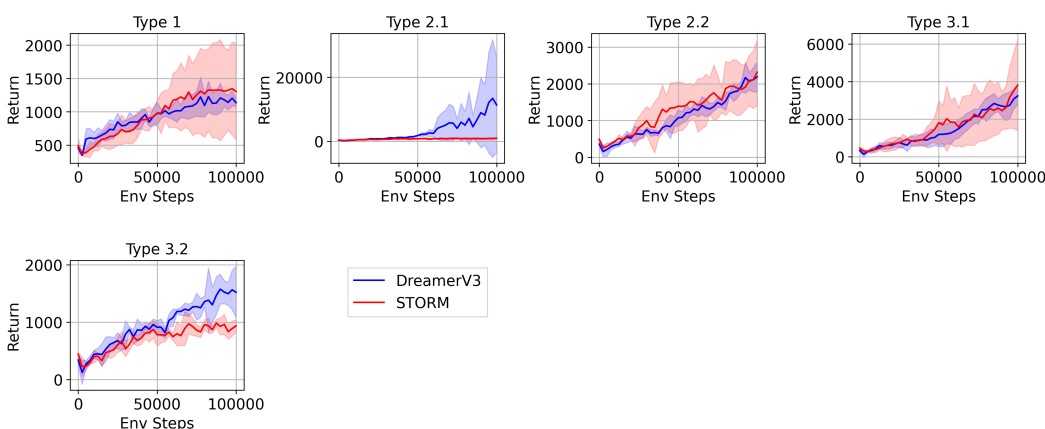

Figure 8: Gopher - learning curves

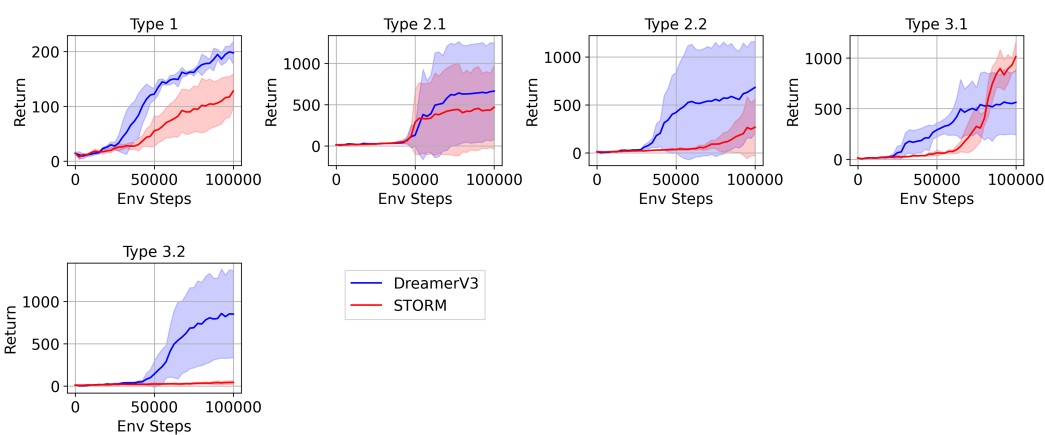

Figure 9: BankHeist - learning curves

## B.3 OVERALL RESULTS FOR EVALUATION RETURN

Table 4: Overall Results for Evaluation Return

| GAME NAME | STOCHASTICITY TYPE | DREAMERV3 | STORM |
|---|---|---|---|
| Breakout | 1 | $8.67 \pm 0.30$ | $9.20 \pm 1.72$ |
| | 2.1 | $13.80 \pm 3.26$ | $16.44 \pm 2.03$ |
| | 2.2 | $14.86 \pm 1.74$ | $20.45 \pm 2.92$ |
| | 3.1 (Default Baseline) | $60.71 \pm 41.89$ | $24.17 \pm 3.55$ |
| | 3.2A | $10.65 \pm 0.41$ | $12.05 \pm 0.89$ |
| | 3.2B | $3.57 \pm 1.43$ | $6.48 \pm 2.50$ |
| | 3.2C | $1.89 \pm 0.49$ | $3.96 \pm 1.42$ |
| Boxing | 1 | $39.32 \pm 6.79$ | $29.48 \pm 15.41$ |
| | 2.1 | $43.00 \pm 31.98$ | $54.69 \pm 20.44$ |
| | 2.2 | $42.21 \pm 35.54$ | $22.44 \pm 3.54$ |
| | 3.1 (Default Baseline) | $84.22 \pm 1.68$ | $86.18 \pm 11.29$ |
| | 3.2A | $86.90 \pm 1.33$ | $85.22 \pm 11.77$ |
| | 3.2B | $61.74 \pm 17.71$ | $45.41 \pm 26.56$ |
| | 3.2C | $56.52 \pm 18.45$ | $52.66 \pm 25.68$ |
| Gopher | 1 | $1137.00 \pm 56.98$ | $1303.53 \pm 724.76$ |
| | 2.1 | $11333.53 \pm 14761.12$ | $950.67 \pm 188.37$ |
| | 2.2 | $2190.87 \pm 407.24$ | $2315.40 \pm 893.32$ |
| | 3.1 (Default Baseline) | $3235.27 \pm 443.51$ | $3811.67 \pm 2431.85$ |
| | 3.2 | $1521.13 \pm 451.45$ | $936.40 \pm 106.83$ |
| BankHeist | 1 | $197.63 \pm 20.76$ | $128.03 \pm 31.41$ |
| | 2.1 | $663.60 \pm 587.85$ | $467.80 \pm 507.74$ |
| | 2.2 | $682.67 \pm 476.90$ | $267.70 \pm 295.86$ |
| | 3.1 (Default Baseline) | $562.30 \pm 320.74$ | $1015.73 \pm 148.43$ |
| | 3.2 | $849.80 \pm 514.15$ | $43.10 \pm 22.85$ |

## B.4  ADDITIONAL DETAILS: TYPE 2.1 ERROR ANALYSIS

We conducted a 1000-step probe in BANKHEIST under Type 2.1 stochasticity. During this probe, we executed a fixed action (action 3) for all 1000 steps and collected:

1. the true environment frames, and

2. the corresponding world-model predictions.

For each sequence, we computed pixel-wise variances and compared:

- Type 3.1 (default, non-stochastic environment—baseline), and

- Type 2.1 (environment with stochastic teleportation events).

Across the 1000-step Type 2.1 run, four teleportation events occurred, producing large variance in the environment-state sequence.

DREAMERV3 RESULTS.

$$\text{Env variance difference } (2.1 - 3.1) \approx 299.10$$
$$\text{Model variance difference } (2.1 - 3.1) \approx 0.0047$$
$$\text{Env–model variance gap (Type 2.1)} \approx 300.34$$

STORM RESULTS.

$$\text{Env variance difference } (2.1 - 3.1) \approx 323.90$$
$$\text{Model variance difference } (2.1 - 3.1) \approx 0.012$$
$$\text{Env–model variance gap (Type 2.1)} \approx 325.21$$

**Interpretation.**  The large environment variance difference between Type 2.1 and Type 3.1 highlights that, under a fixed action sequence, the baseline environment (Type 3.1) exhibits extremely low variance, whereas the stochastic environment (Type 2.1) exhibits dramatically higher variance due to action-independent teleportation events. This demonstrates how sensitive the true environment dynamics are to exogenous stochasticity.

In contrast, the world models fail to capture both the magnitude and the structure of this variance. Even when the true environment undergoes large, abrupt, action-independent deviations (e.g., teleportation), the model predictions remain nearly unchanged from the deterministic baseline. Thus, while the environment distribution widens substantially under Type 2.1, the learned world models continue to produce near-deterministic, low-variance prediction streams, indicating that they severely underestimate the true stochasticity.

## B.5 ADDITIONAL DETAILS: TYPE 3.2 ERROR ANALYSIS

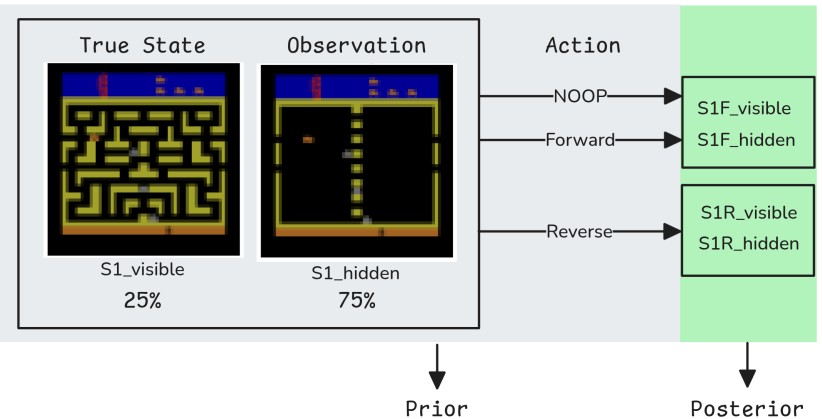

Figure 10: Detailed results from the BankHeist Type 3.2 partial observability probe. Each row corresponds to one of six carefully selected cases, showing the start state visibility, next state visibility, negative log-likelihood (NLL), and KL divergence for both DreamerV3 and STORM.

Table 5: Analysis on BankHeist (Type 3.2) - DreamerV3

| Prior Obs | Metrics | Action — Posterior Observations (Visible/Hidden) | | | | | |
|---|---|---|---|---|---|---|---|
| | | NOOP (0) S1F_visible | Forward (3) S1F_visible | Reverse (4) S1R_visible | NOOP (0) S1F_hidden | Forward (3) S1F_hidden | Reverse (4) S1R_hidden |
| S1_visible | - log d | 41.72 | 38.43 | 44.73 | 19.13 | 21.56 | 29.81 |
| | KL div | 25.52 | 20.13 | 24.71 | 6.40 | 7.17 | 12.78 |
| S1_hidden | - log d | 41.36 | 38.74 | 45.14 | 25.15 | 22.63 | 29.25 |
| | KL div | 20.99 | 20.47 | 24.57 | 10.72 | 7.05 | 11.81 |

Table 6: Analysis on BankHeist (Type 3.2) - STORM

| Prior Obs | Metrics | Action — Posterior Observations (Visible/Hidden) | | | | | |
|---|---|---|---|---|---|---|---|
| | | NOOP (0) S1_visible | Forward (3) S1F_visible | Reverse (4) S1R_visible | NOOP (0) S1_hidden | Forward (3) S1F_hidden | Reverse (4) S1R_hidden |
| S1_visible | - log p | 33.53 | 21.72 | 28.65 | 19.81 | 11.29 | 21.86 |
| | KL div | 115.63 | 116.00 | 114.13 | 114.94 | 115.47 | 113.79 |
| S1_hidden | - log p | 28.60 | 18.50 | 22.58 | 18.43 | 12.61 | 16.21 |
| | KL div | 115.55 | 115.49 | 115.05 | 115.03 | 115.19 | 114.73 |

### B.6 COMPUTE RESOURCES USED

The experiment runs were executed in several types of GPUs like A40, A100 and H100 depending on availability. Each node atleast had 32 vCPU and 50GB RAM. On GPUs with large memory, mulitple runs were executed.

DreamerV3 and STORM took around 24 hours and 12 hours respectively per run (training & evaluation) per seed when running on single GPU.

## C INFORMATION-THEORETIC LEVERS

Information-theoretic measures provide quantitative levers to diagnose how stochasticity affects learning and planning:

**Action channel capacity.** For action-dependent noise, controllability is reduced. The effective capacity is measured by $I(A; \tilde{A} \mid S)$, which quantifies how much of the intended action $A$ survives corruption into the executed action $\tilde{A}$.

**Predictive information of dynamics.** For action-independent randomness, the predictive structure is measured by $I((S_t, A_t); S_{t+1})$, reflecting how much the next state depends on the current state-action pair. Under drift, temporal changes in this quantity indicate shifts in environment regularity.

**Representation sufficiency.** A latent $Z_t$ should act as a sufficient statistic for planning. Ideally, $I(Z_t; S_t)$ is maximized, while $I(Z_t; O_t)$ remains bounded, ensuring that $Z_t$ captures hidden states rather than surface-level noise, consistent with bisimulation invariance.

**Aliasing quantification.** In partially observable settings, the observation-state information gap can be written as $I(S_t; O_t) - I(S_t; O_t \mid A_t)$, capturing residual uncertainty after conditioning on actions. This disentangles sensor noise from genuine state ambiguity.

**Risk-sensitive planning.** Robust planning can be viewed through an information lens: risk-sensitive objectives such as Conditional Value at Risk (CVaR) optimize not the mean return but lower quantiles, effectively re-weighting information from rare but catastrophic outcomes.

# D  ALL IMPLEMENTED STOCHASTICITY MODES

We define stochasticity modes along four Atari environments (Breakout, Boxing, Gopher, BankHeist), and the set of cropping modes are common to all games.

COMMON CROPPING MODES (ALL GAMES)

- Mode 0: No crop
- Mode 1: Left — Crop the left half of the observation
- Mode 2: Right — Crop the right half of the observation
- Mode 3: Top — Crop the top half of the observation
- Mode 4: Bottom — Crop the bottom half of the observation
- Mode 5: Random circular mask — Randomly mask a circular region of the observation

BREAKOUT

**Action-independent random**

- 0: none
- 1: block hit cancel (reward unchanged)
- 2: block hit cancel (reward set to 0)
- 3: regenerate a randomly chosen hit block

**Partial observation (blackout)**

- 0: none
- 1: all
- 2: blocks
- 3: paddle
- 4: score
- 5: ball_missing_top
- 6: ball_missing_middle
- 7: ball_missing_bottom
- 8: blocks_and_paddle
- 9: blocks_and_score
- 10: ball_missing_top_and_bottom
- 11: ball_missing_all

**Partial observation - RAM modification**

- 0: none
- 1: nus_pattern (blocks RAM)
- 2: ball_hidden

BOXING

**Action-independent random**

- 0: none
- 1: colorflip (swap player/enemy colors)
- 2: hit cancel (revert score; reward set to 0)

- 3: displace to corners (swap player/enemy positions)

**Partial observation (blackout)**

- 0: none
- 1: all
- 2: left boxing ring
- 3: right boxing ring
- 4: full boxing ring
- 5: enemy score
- 6: player score
- 7: enemy+player score
- 8: clock
- 9: enemy+player score+clock

**Partial observation - RAM modification**

- 0: none
- 1: hide boxing ring
- 2: hide enemy
- 3: hide player

GOPHER

**Action-independent random**

- 0: none
- 1: hole doesn't close (fill cancel; reward unchanged)
- 2: hole doesn't close (fill cancel; reward set to 0)
- 3: randomly remove one visible carrot (once per reset)

**Partial observation (blackout)**

- 0: none
- 1: all
- 2: gopher attack (both sides)
- 3: left gopher attack
- 4: right gopher attack
- 5: underground full (before-dug color)
- 6: underground full offset (before-dug color)
- 7: underground row 0 (before-dug)
- 8: underground row 0 (dug color)
- 9: underground row 1 (before-dug)
- 10: underground row 1 (dug color)
- 11: underground row 2 (before-dug)
- 12: underground row 2 (dug color)
- 13: underground row 3 (before-dug)
- 14: underground row 3 (dug color)
- 15: farmer (full)

- 16: farmer below nose
- 17: duck fly
- 18: score

**Partial observation - RAM modification**

- 0: none
- 1: hide left carrot
- 2: hide middle carrot
- 3: hide right carrot
- 4: hide all carrots
- 5: hide seed

BANKHEIST

**Action-independent random**

- 0: none
- 1: dropped bomb is a dud
- 2: fuel leaks (per city, once per episode)
- 3: switch city mid-way (teleport)
- 4: bank empty (reward suppressed when bank→police transition detected)

**Partial observation (blackout)**

- 0: none
- 1: all
- 2: city walls (all)
- 3: top city wall
- 4: left city wall
- 5: bottom city wall
- 6: right city wall
- 7: left and right city walls together
- 8: fuel region
- 9: lives region
- 10: score region

**Partial observation - RAM modification**

- 0: none
- 1: hide robber's car
- 2: hide change in fuel (always full)
- 3: hide city blocks
- 4: blend city blocks and wall (background color)
- 5: hide banks (when currently a bank)
- 6: hide police (when currently police)

CONCEPT DRIFT USAGE

All *partial observation*, *action-independent*, and *action-dependent* modes can also be used as a **second concept** in a concept drift setting, enabling controlled evaluation of robustness to non-stationary environments.

# E  THE USE OF LARGE LANGUAGE MODELS (LLMs)

We made use of large language models (LLMs) to assist with selected aspects of this work. Specifically, LLMs were employed to improve the clarity and flow of writing, to summarize and condense long paragraphs during manuscript preparation, and to generate code snippets for repetitive components of the implementation. All outputs from the LLMs were carefully reviewed, validated, and edited by the authors to ensure accuracy and correctness.

