# OpenReview forum: "STORI: A Benchmark and Taxonomy for Stochastic Environments"
_ICLR.cc/2026/Conference — Submitted to ICLR 2026_

### Official Review · Reviewer_Uy4R · 2025-10-26

**Soundness:** 1
**Presentation:** 3
**Contribution:** 3
**Rating:** 2
**Confidence:** 4

**Summary:**

This paper introduces STORI, a new benchmark for testing how robust RL agents are in stochastic environments that contain uncertainty. The authors argue that current benchmarks, like the Atari 100k, are too deterministic, which hides how brittle RL agents are and doesn't help when applied in the real world. Their contribution consists of: (1) A new five-type taxonomy of stochasticity, and (2) The STORI framework, which adds these types of stochasticity as wrappers on top of four Atari games. They test two modern model based RL agents, DreamerV3 and STORM, and find they can't handle these new kinds of uncertainty. The paper's most compelling contribution is the targeted probing of the world models. These probes reveal several fundamental flaws, for example, that the learned models underestimate the true environmental variance by a very large factor (~300x).

**Strengths:**

1. The work tackles a big, known gap between how RL agents do in clean simulators and how they perform in the real world. Trying to build a systematic benchmark for this is a good, well motivated, goal.
2. The proposed five-type taxonomy of stochasticity is a clear and helpful way to organize the problem, which seems like it covers the important aspects of the problem.
3. The paper's best part is the analysis in probe analysis showing that SOTA world models miss the environment's built-in randomness (the ~300x variance underestimation claim). The "clear-start" vs. "obscured-start" test is also a clever way to test how robust the agent's internal beliefs are.

**Weaknesses:**

1. The biggest problem with this paper is that its "benchmark for RL" only tests two agents from one single family (model-based RL). The conclusions are all about "world models," but they're written up as if they apply to all of RL. We have no idea how model-free agents, the other main approach, would perform. Do they also break, or is this a specific problem for MBRL?
The paper abstract claims that STORI “...is a benchmark that systematically incorporates diverse stochastic effects and enables rigorous evaluation of RL techniques under different forms of uncertainty.”
This is misleading, as the results in the paper completely ignore an entire family of algorithms. The authors need to fix this by including SOTA MF baselines that perform well on the Atari 100k standard benchmark. What is the reason for the omission?
2. The paper only used 3 seeds for all experiments. This could be enough for a general RL paper, but for a benchmark that's about stochasticity, this might not be enough. For example, re table 5, the authors state in the text “DreamerV3 shows positive NLL values”. But the value in the table is  1.15±2.46, the mean is within the std of the result! This heavily questions the conclusions drawn from these experiments.

**Questions:**

Please address the points in the Weaknesses section. In addition:

1. The probe in Section 5.2.1 (variance underestimation) is your most interesting finding. Can you give more detail on how the ~300x variance difference was calculated? Is this an average over the 1000-step probe, a final-step measurement, or just one example?

---

> ### Author Response · Authors · 2025-12-03
>
> Reviewer Comment:
>
> The biggest problem with this paper is that its "benchmark for RL" only tests two agents from one single family (model-based RL). The conclusions are all about "world models," but they're written up as if they apply to all of RL. We have no idea how model-free agents, the other main approach, would perform. Do they also break, or is this a specific problem for MBRL? The paper abstract claims that STORI “...is a benchmark that systematically incorporates diverse stochastic effects and enables rigorous evaluation of RL techniques under different forms of uncertainty.” This is misleading, as the results in the paper completely ignore an entire family of algorithms. The authors need to fix this by including SOTA MF baselines that perform well on the Atari 100k standard benchmark. What is the reason for the omission?
>
>
> Response:
>
> Thank you for the question regarding the comparison to model-free methods. Our choice to focus on DreamerV3 and STORM was driven primarily by practical constraints of the experimental setting.
>
> Our study evaluates robustness under a 100K-interaction regime, which is designed to stress test sample-efficient algorithms. Because we evaluate multiple types and magnitudes of stochastic perturbations across four games, the total experimental cost is substantial. This limited our ability to include a broader range of algorithms in the initial submission.
>
> Most model-free methods are not competitive in 100K steps:
> Many standard model-free Atari agents (e.g., PPO, Rainbow, IQN etc.) typically require millions of environment frames to reach reasonable performance and thus are not directly comparable in the 100K-step setting used for DreamerV3 and STORM. Including such methods would provide little meaningful signal regarding robustness because they underperform even in the non-stochastic setting at this sample budget.
>
> There do exist promising fast-learning model-free approaches such as BBF (“Bigger, Better, Faster: Human Level Atari with Human Level Efficiency”). However, BBF’s public implementation is in JAX, and preliminary attempts to integrate it into our PyTorch-based evaluation pipeline introduced some slowdowns and incompatibilities during testing.
>
> We agree that adding sample-efficient model-free baselines would strengthen the generality of the conclusions. We are currently benchmarking several such algorithms in our framework and plan to include these results in an updated version of the paper.
>
> Overall, while our current experiments focus on two state-of-the-art model-based agents, the design choices were motivated by fairness and feasibility in the 100K-step regime. We appreciate your suggestion, and additional model-free comparisons are actively being incorporated.

---

> ### Author Response · Authors · 2025-12-03
>
> Reviewer Comment:
>
> The paper only used 3 seeds for all experiments. This could be enough for a general RL paper, but for a benchmark that's about stochasticity, this might not be enough. For example, re table 5, the authors state in the text “DreamerV3 shows positive NLL values”. But the value in the table is 1.15±2.46, the mean is within the std of the result! This heavily questions the conclusions drawn from these experiments.
>
> Response:
>
> Thank you for pointing this out. As stated in “B.4 Additional Details: Type 3.2 Error Analysis” in the appendix, the ΔNLL evaluation was conducted on six carefully selected cases using a checkpoint from one of the seed models. Therefore, the variance observed in Table 5 is not directly attributable to the use of three seeds in the main results, but rather to the limited number of cases included in the error analysis itself.
>
> We apologize for the confusion and will clarify this distinction more explicitly in the revised version.

---

> ### Author Response · Authors · 2025-12-03
>
> Reviewer Comment:
>
> The probe in Section 5.2.1 (variance underestimation) is your most interesting finding. Can you give more detail on how the ~300x variance difference was calculated? Is this an average over the 1000-step probe, a final-step measurement, or just one example?
>
> Response:
>
> Thank you for highlighting this point. We will expand the explanation of the variance-underestimation in the revised version.
>
> To clarify, the “∼300×” variance difference is computed from a 1000-step probe in BankHeist under Type 2.1 stochasticity. In this probe, we executed a fixed action (action 3) for 1000 consecutive steps and collected
>
> (i) true environment frames, and
> (ii) world-model predictions.
>
> We then computed pixel-wise variances for each sequence and compared:
>
>     -Type 3.1 (default, non-stochastic environment — baseline)
>
>     -Type 2.1 (with stochastic teleportation events)
>
> During the 1000-step Type 2.1 run, four teleportation events occurred, producing very large variance in the environment state sequence.
> ```
>    For DreamerV3:
>    Env variance difference (2.1 − 3.1): ≈ 299.10
>    Model variance difference (2.1 − 3.1): ≈ 0.0047
>    Env–model variance gap (Type 2.1): ≈ 300.34
> ```
> ```
>    For STORM:
>    Env variance difference (2.1 − 3.1): ≈ 323.90
>    Model variance difference (2.1 − 3.1): ≈ 0.012
>    Env–model variance gap (Type 2.1): ≈ 325.21
> ```
> Importantly, the environment variance difference between Type 2.1 and Type 3.1 highlights that, under the same fixed sequence of actions, the baseline environment (Type 3.1) produces extremely low variance, whereas the stochastic environment (Type 2.1) produces dramatically higher variance due to the teleportation events. This illustrates how sensitive the true environment dynamics are to action-independent stochastic events for this environment.
>
> The world models fail to capture both the magnitude and the structure of this variance. Even when the true environment exhibits large, abrupt, action-independent deviations (e.g., teleportation events), the model predictions remain almost unchanged compared to the deterministic baseline (Type 3.1). In other words, while the environment distribution widens dramatically under Type 2.1, the learned world models continue to produce a near-deterministic, low-variance prediction stream, indicating that they substantially underestimate the true stochasticity.
>
> We will include this clarification and a more detailed explanation of the probe computation in the revised version of the manuscript.

---

### Official Review · Reviewer_mPT5 · 2025-10-31

**Soundness:** 3
**Presentation:** 2
**Contribution:** 3
**Rating:** 4
**Confidence:** 4

**Summary:**

This paper introduces STORI, a benchmark suite for studying stochasticity in reinforcement learning environments under different forms of uncertainty. The authors formalize a taxonomy composed of five types of uncertainty (action-dependent
noise, action-independent randomness, concept drift, representation learning challenges,
and missing state information) and provide wrapper implementations on top of existing RL environments including these stochastic events. To evaluate the framework, authors conduct experiments with two model-based RL algorithms (DreamerV3, STORM) to illustrate how standard world models behave under each type of stochasticity. They also provide an open-source repository to test the framework.

**Strengths:**

- **Timely and practically relevant problem:** The framework explicitly targets a crucial aspect of world models and RL agents which are influenced by real-world uncertainty.
- **Sound conceptual formalization:** The study of the different types of uncertainty as a unified integration of existing perspectives is well-motivated and tied to concrete environment mechanisms (e.g., sticky actions, time-varying transition kernels, observation aliasing).
- **Accessible and modular benchmark:** The wrappers appear easy to integrate, allowing systematic testing of RL algorithms under controlled perturbations.
- **Interesting evaluations:** The analysis of aleatoric vs. epistemic/aliasing effects via the law of total variance is conceptually clean and useful.

**Weaknesses:**

- **Mathematical underspecification:** Overall the mathematical notation lacks formality. Several formulas use an undefined divergence D(·||·) (e.g., for measuring concept drift) and do not specify how this is aggregated over states and actions. The notation for the corruption channel alternates between C(ã|a) and C(ã|s,a) with no explanation. Equation 4 shows the Bayes filter update as a proportional relation but does not include or mention the normalization term. The observation aliasing statement (“O(o|s_i)=O(o|s_j)=1”) is mathematically incorrect: it should read O(·|s_i) = O(·|s_j) to indicate identical observation distributions.

- **Relativity of results:** It is difficult to assess the relative performance of the evaluated algorithms since results are not compared against a clearly defined baseline (e.g., the non-perturbed environment).

- **Learning curves:** Some learning curves in the appendix do not reach stability, raising concerns about premature training. Listing algorithm hyperparameters in the appendix would improve reproducibility.

- **Editorial and referencing issues:** Equations 12/14 and Table 3 are inconsistently referenced or missing from the text, suggesting incomplete editing. Several cross-references (e.g., between definitions in section 3.2 and results) are broken or misnumbered.

- **Difficult to follow:** While until section 4 the paper is easy to follow and clearly written, it becomes a bit cumbersome in the experiment section. The evaluation is inconsistent across uncertainty types (for example the analysis on type 2.1 is not done on type 2.2). Also, to understand that even unmentioned types have been evaluated (like type 1) is necessary to read the appendix. At the same time, tables and plots hinder the understanding since they are weakly commented and sometimes misplaced, i.e., they are sometimes placed far from their discussion, making navigation cumbersome (e.g., Figure 1b is at page 1 but described at page 7).

**Questions:**

- At lines 336 and 353, authors mention a baseline and comment the performance drop of Breakout with uncertanty type 2.1 of 15% wrt a baseline. Though is not really clear what this baseline actually is. Please, describe it a little bit.
- What is the intuition behind the analysis of the error caused by type 2.1 whose relative variance is compared with the one of type 3.1 and only with that? Why an equivalent analysis is not provided for type 2.2, for example?
- Does the framework allow to combine multiple uncertainty types at the same time?

---

> ### Author Response · Authors · 2025-12-03
>
> Reviewer Comment:
>
> Relativity of results: It is difficult to assess the relative performance of the evaluated algorithms since results are not compared against a clearly defined baseline (e.g., the non-perturbed environment).
>
> Response:
>
> Thank you for raising this important point. We apologize for not making the role of the non-perturbed (Type 3.1) environment clearer as the reference baseline for interpreting all results.
>
> How baseline comparability is intended in our setup?:
>
>     In STORI, Type 3.1 corresponds directly to the standard, non-perturbed ALE environment. For all stochasticity types where the reward semantics remain unchanged, performance in Type 3.1 serves as the baseline against which degradation can be meaningfully interpreted. Under this criterion:
>
>      - All stochasticity types for Boxing and BankHeist preserve reward meaning and can be compared directly to their Type 3.1 baselines.
>
>      - For Breakout and Gopher, all stochasticity types except Type 2.1 preserve reward semantics and thus allow valid baseline comparisons.
>
> Why some stochasticity types do not support baseline comparison?:
>
>     Certain perturbations alter the reward-generation process itself. In Type 2.1 for Breakout and Gopher, for example, events such as successful hits may be stochastically canceled, lowering the maximum achievable score even for an optimal agent. Because these settings shift the underlying reward distribution, comparing them directly to the non-perturbed baseline would be misleading.
>
> Why our reported results focus on within-type comparisons?:
>
>     Due to these inconsistencies, the main paper emphasizes algorithmic comparisons within each stochastic environment, which offers a more reliable and fair assessment of robustness. However, we acknowledge that the lack of explicit discussion made it difficult to interpret relative performance.
>
> We will revise the manuscript to clearly define the Type 3.1 environment as the baseline, explain when baseline comparisons are valid, and include all appropriate baseline-anchored comparisons wherever reward consistency is preserved.

---

> ### Author Response · Authors · 2025-12-03
>
> Reviewer Comment:
>
> Learning curves: Some learning curves in the appendix do not reach stability, raising concerns about premature training. Listing algorithm hyperparameters in the appendix would improve reproducibility.
>
> Response:
>
> Thank you for the helpful comment. Our study evaluates robustness under a 100K-steps regime, which is intentionally chosen to stress-test the sample efficiency of the evaluated algorithms. Because we examine multiple types and magnitudes of stochastic perturbations across four games, the total experimental cost is substantial, which limited our ability to extend training runs or include a broader set of algorithms in the initial submission.
>
> Within this 100K-step constraint, some learning curves in the appendix do not reach full stability. This behavior is expected and acceptable in our setup, as the goal is to assess early-learning robustness rather than long-horizon convergence.
>
> We agree that listing all algorithm hyperparameters would improve reproducibility, and we will include a complete hyperparameter table in the revised version.

---

> ### Author Response · Authors · 2025-12-03
>
> Reviewer Comment:
>
> Editorial and referencing issues:
>
>     Equations 12/14 and Table 3 are inconsistently referenced or missing from the text, suggesting incomplete editing. Several cross-references (e.g., between definitions in section 3.2 and results) are broken or misnumbered.
>
>
> Difficult to follow:
>
>     While until section 4 the paper is easy to follow and clearly written, it becomes a bit cumbersome in the experiment section. The evaluation is inconsistent across uncertainty types (for example the analysis on type 2.1 is not done on type 2.2). Also, to understand that even unmentioned types have been evaluated (like type 1) is necessary to read the appendix. At the same time, tables and plots hinder the understanding since they are weakly commented and sometimes misplaced, i.e., they are sometimes placed far from their discussion, making navigation cumbersome (e.g., Figure 1b is at page 1 but described at page 7).
>
>
> Response:
>
> Thank you for bringing these issues to our attention. We apologize for the editing and referencing errors, including the inconsistent citations of Equations 12/14, Table 3, and several broken or misnumbered cross-references. We will correct all of these in the revised version to ensure smooth navigation and consistency throughout the manuscript.
>
> Regarding the evaluation section, we acknowledge that the presentation became harder to follow. The uneven appearance of analyses across stochasticity types stems from a typo in the heading for the Type 2.1 error analysis; this section actually includes the analysis for Type 2.2 as well. We will fix this labeling issue and reorganize the section so that the evaluation for each stochasticity type is clearly delineated and easy to locate.

---

> ### Author Response · Authors · 2025-12-03
>
> Comment:
> Does the framework allow to combine multiple uncertainty types at the same time?
>
> Response:
>
> While it is possible, with some modifications to the current STORI implementation, to combine multiple stochasticity modes and study their interactions, this is currently outside the scope of the present paper. We believe it's a valuable direction for future work.

---

> ### Author Response · Authors · 2025-12-03
>
> Reviewer Comment:
>
> Mathematical underspecification: Overall the mathematical notation lacks formality. Several formulas use an undefined divergence D(·||·) (e.g., for measuring concept drift) and do not specify how this is aggregated over states and actions. The notation for the corruption channel alternates between C(ã|a) and C(ã|s,a) with no explanation. Equation 4 shows the Bayes filter update as a proportional relation but does not include or mention the normalization term. The observation aliasing statement (“O(o|s_i)=O(o|s_j)=1”) is mathematically incorrect: it should read O(·|s_i) = O(·|s_j) to indicate identical observation distributions.
>
> Response:
>
> Thank you for raising these important issues. We acknowledge that several parts of the mathematical specification were underspecified or inconsistently notated, including the definition of the divergence D(⋅∥⋅), the notation for the corruption channel, the missing normalization term in the Bayes filter update, and the incorrect expression for observation aliasing. We will revise the paper to formalize all mathematical definitions, ensure consistent notation, and correct the identified errors in the camera-ready version.

---

### Official Review · Reviewer_cTXg · 2025-10-31

**Soundness:** 2
**Presentation:** 3
**Contribution:** 2
**Rating:** 2
**Confidence:** 4

**Summary:**

The authors introduce STORI, a benchmark designed to simulate various types of environmental stochasticity, and propose a five-type taxonomy to categorize such uncertainties. They conduct experiments on two state-of-the-art model-based RL algorithms, DreamerV3 and STORM, and show that these methods suffer significant performance degradation when stochasticity is introduced. The results highlight that existing world models tend to underestimate environmental variance, are sensitive to action corruption, and struggle under partial observability.

**Strengths:**

The main strengths of this paper lie in its clear motivation and systematic approach to a well-recognized problem. The proposed taxonomy provides a useful conceptual framework for discussing stochasticity, and the benchmark implementation can help the community evaluate algorithms under diverse types of uncertainty. Moreover, the empirical analysis is thorough and provides valuable insights into the weaknesses of existing model-based RL methods when facing stochastic dynamics. The public release of the benchmark and code is also a strong practical contribution that may encourage further research in this direction.

**Weaknesses:**

The experiments primarily evaluate DreamerV3 and STORM, which are not explicitly designed for stochastic environments. While the performance drop observed in these methods is interesting, it is also expected, and readers might be more curious about how adversarial training, domain randomization, or uncertainty-aware approaches perform in comparison. Including such baselines would strengthen the empirical findings and make the results more informative for practitioners aiming to improve robustness.

In terms of novelty, although the proposed STORI benchmark is more systematic and comprehensive than previous attempts, similar studies have explored stochastic perturbations in RL environments. The authors should better position their work relative to these efforts, clarifying what aspects of coverage, taxonomy, or reproducibility truly distinguish STORI from prior benchmarks.

The definition of the first type of stochasticity, in which the environment repeats the previous action with some probability, captures one form of action-dependent noise. However, it might not fully represent the broader class of real-world cases. For example, gradual drift in actuator response, changes in friction, or variations in torque output due to power fluctuations could also be categorized under this type. Expanding or at least discussing such scenarios would make the taxonomy more realistic and convincing.

Finally, the presentation of results can be improved. In the current tables, the authors mainly report the relative performance difference between stochastic and non-stochastic settings. It would be clearer to also include the absolute performance scores for each method, as this would help readers interpret the significance of the degradation. For instance, if DreamerV3 consistently outperforms STORM in the base environment, a larger performance drop under stochasticity may be less surprising. Presenting all results in well-organized summary tables would also make the paper easier to follow.

**Questions:**

Would it be more informative to include comparisons with methods that explicitly address uncertainty, such as adversarial or domain randomization approaches?

Have the authors analyzed whether different forms of stochasticity interact, i.e., whether combining multiple types leads to compounding effects on performance?

---

> ### Author Response · Authors · 2025-12-03
>
> Reviewer Comment:
>
> The experiments primarily evaluate DreamerV3 and STORM, which are not explicitly designed for stochastic environments. While the performance drop observed in these methods is interesting, it is also expected, and readers might be more curious about how adversarial training, domain randomization, or uncertainty-aware approaches perform in comparison. Including such baselines would strengthen the empirical findings and make the results more informative for practitioners aiming to improve robustness.
>
> Would it be more informative to include comparisons with methods that explicitly address uncertainty, such as adversarial or domain randomization approaches?
>
> Response:
>
> Thank you for the helpful suggestion. In this work, our initial goal is to first evaluate how standard state-of-the-art RL agents (DreamerV3 and STORM), which although are not explicitly designed for stochastic settings, behave under the diverse uncertainty conditions introduced by STORI. We agree that comparing against uncertainty-aware, adversarially trained, or domain-randomized approaches would provide valuable additional insights. We see this as a natural extension of the benchmark, and we plan to explore these classes of robustness-oriented methods in future work.

---

> ### Author Response · Authors · 2025-12-03
>
> Reviewer Comment:
>
> The definition of the first type of stochasticity, in which the environment repeats the previous action with some probability, captures one form of action-dependent noise. However, it might not fully represent the broader class of real-world cases. For example, gradual drift in actuator response, changes in friction, or variations in torque output due to power fluctuations could also be categorized under this type. Expanding or at least discussing such scenarios would make the taxonomy more realistic and convincing.
>
> Response:
>
> Thank you for this helpful suggestion. To clarify, in environments with action-dependent intrinsic stochasticity, the environment may replace the agent’s chosen action with a different one. The random-action replacement mechanism we use (including the “sticky actions” variant) is intended as an example instantiation of this broader category, not an exhaustive definition. We apologize for the lack of clarity in the original description of the first stochasticity type.
>
> Regarding the scenarios you mentioned, gradual drift in actuator response could indeed be viewed as an interesting mixture of Type 1 (action-dependent stochasticity) and Type 2.2 (concept drift). Similarly, changes in friction or variations in torque output due to power fluctuations align well with Type 1, as they effectively alter the executed action relative to the agent’s intended command through a stochastic or noisy transformation.
>
> This suggestion is very helpful for improving the clarity and realism of the taxonomy, and we will incorporate this refinement into the revised version.

---

> ### Author Response · Authors · 2025-12-03
>
> Reviewer Comment:
>
> Finally, the presentation of results can be improved. In the current tables, the authors mainly report the relative performance difference between stochastic and non-stochastic settings. It would be clearer to also include the absolute performance scores for each method, as this would help readers interpret the significance of the degradation. For instance, if DreamerV3 consistently outperforms STORM in the base environment, a larger performance drop under stochasticity may be less surprising. Presenting all results in well-organized summary tables would also make the paper easier to follow.
>
> Response:
>
> Thank you for the thoughtful suggestion. To clarify, in the paper we focus on comparing the performance of different algorithms within each stochastic environment, which is why the main tables emphasize relative degradation under different stochasticity types. We do, however, provide a detailed table of absolute numerical scores in the appendix, and Type 3.1 in our taxonomy corresponds directly to the baseline environment used for those comparisons.
>
> We apologize for any confusion caused by the current presentation and will revise the text and tables to make this relationship clearer and ensure that readers can easily interpret both absolute performance and relative degradation.

---

> ### Author Response · Authors · 2025-12-03
>
> Reviewer Comment:
> Have the authors analyzed whether different forms of stochasticity interact, i.e., whether combining multiple types leads to compounding effects on performance?
>
> Response:
>
> Thank you for the suggestion. While it is possible, with some modifications to the current STORI implementation, to combine multiple stochasticity modes and study their interactions, this is currently outside the scope of the present paper. We agree that analyzing whether different forms of stochasticity compound or interact in non-trivial ways is an important and valuable direction for future work.

---

### Official Review · Reviewer_PHdh · 2025-11-01

**Soundness:** 3
**Presentation:** 3
**Contribution:** 2
**Rating:** 4
**Confidence:** 4

**Summary:**

This paper presents STORI (STOchastic-ataRI), a new benchmark and taxonomy developed to systematically assess the robustness of Reinforcement Learning (RL) algorithms under stochastic environmental conditions. The central contribution lies in the STORI benchmark, which introduces a diverse range of stochastic effects across four classic Atari environments: Breakout, Boxing, Gopher, and BankHeist.
The authors further propose a comprehensive five-category taxonomy to characterize different types of environmental stochasticity.
Through experiments evaluating two state-of-the-art MBRL algorithms, DreamerV3 and STORM, the study uncovers systematic weaknesses in how these models handle uncertainty and non-determinism.

**Strengths:**

1. STORI offers a systematic framework that enables fine-grained introduction and control of diverse stochastic effects in standard Atari environments.

2. The evaluation extends beyond aggregate performance metrics by analyzing concrete failure modes, such as variance underestimation and state aliasing, to diagnose the root causes of algorithmic weaknesses.

**Weaknesses:**

1. The evaluation focuses solely on two model-based algorithms, DreamerV3 and STORM, which constrains the generalizability of the conclusions regarding robustness across the broader RL landscape.
2. The benchmark focuses on four selected Atari environments, but the paper does not discuss how these tasks were chosen. As a result, potential researcher bias in task selection, and its impact on benchmark representativeness, remains unaddressed.
3. A key limitation lies in cross-type comparability, the varying intensity and nature of stochastic effects make it difficult to draw direct conclusions about algorithm robustness across all categories.

**Questions:**

1. Are current model-free approaches more or less resilient to stochastic perturbations compared to model-based methods?
2. The paper indicates that DreamerV3’s world model exhibits brittleness under Type 3.2 conditions (Missing State Variables). Would increasing the model’s history-tracking capacity mitigate this limitation, or is the weakness more fundamentally rooted in its belief state update mechanism?

---

> ### Author Response · Authors · 2025-11-28
>
> Thank you for the question regarding the comparison to model-free methods. Our choice to focus on DreamerV3 and STORM was driven primarily by practical constraints of the experimental setting.
>
> - Our study evaluates robustness under a 100K-interaction regime, which is designed to stress test sample-efficient algorithms. Because we evaluate multiple types and magnitudes of stochastic perturbations across four games, the total experimental cost is substantial. This limited our ability to include a broader range of algorithms in the initial submission.
>
> - Many standard model-free Atari agents (e.g., PPO, Rainbow, IQN etc.) typically require millions of environment frames to reach reasonable performance and thus are not directly comparable in the 100K-step setting used for DreamerV3 and STORM. Including such methods would provide little meaningful signal regarding robustness because they underperform even in the non-stochastic setting at this sample budget. There do exist promising fast-learning model-free approaches such as BBF (“Bigger, Better, Faster: Human Level Atari with Human Level Efficiency”). However, BBF’s public implementation is in JAX, and preliminary attempts to integrate it into our PyTorch-based evaluation pipeline introduced some slowdowns and incompatibilities during testing.
>
> - We agree that adding sample-efficient model-free baselines would strengthen the generality of the conclusions. We are currently benchmarking several such algorithms in our framework and plan to include these results in an updated version of the paper.
>
> Overall, while our current experiments focus on two state-of-the-art model-based agents, the design choices were motivated by fairness and feasibility in the 100K-step regime. We appreciate your suggestion, and additional model-free comparisons are actively being incorporated.

---

> ### Author Response · Authors · 2025-11-28
>
> Thank you for highlighting the need for a clearer explanation of our environment selection. We apologize for the lack of explicit justification in the main paper and clarify our criteria below.
>
> - Rationale for selecting the four Atari 100K environments
>
>     We chose Breakout, Boxing, Gopher, and BankHeist based on their optimality classifications in Lim et al., 2025. Breakout, Boxing, and Gopher are Agent-Optimal tasks, whereas BankHeist is Human-Optimal. Our primary goal was to study robustness in settings where agents already achieve strong performance under default conditions, making performance degradation under stochasticity easier to observe and quantify.
>     We included one Human-Optimal task (BankHeist) to obtain a preliminary understanding of how added stochasticity interacts with environments where agents are known to perform substantially worse than humans even in the canonical setting.
>
>     Additionally, this set provides diverse action-space sizes: Breakout (4 actions), Gopher (8), and Boxing/BankHeist (18 each), helping ensure that our robustness observations are not tied to a particular action-space complexity.
>
> - Selection of tasks for individual stochasticity types
>
>
>     As noted in the paper, the assignment of stochasticity types to specific environments was mostly random, with the exception of type 2.2 (concept drift). For concept drift, we intentionally selected concept 2 to be one of the other stochasticity types, allowing for a direct and controlled comparison between “pure” stochasticity and its drifting counterpart.
>
> - Reason for multiple stochasticity variants per environment
>
>     We provide several parameter settings for each stochasticity type and each game so that users of the benchmark can match the perturbation regime most relevant to the real-world stochasticity their algorithms aim to handle. The broader range helps ensure that the benchmark remains useful across different research focuses and robustness requirements.
>
> We appreciate your feedback and will incorporate a clear and explicit description of these selection criteria and their motivation in the camera-ready version.

---

> ### Author Response · Authors · 2025-11-28
>
> Thank you for raising an important point in Weakness (3). We apologize for not explaining the limits of cross-type comparability more explicitly in the main paper. We clarify the intended comparability structure below.
>
> - When cross-type comparisons within an environment are valid:
>
>    Cross-type comparability is feasible when the semantic meaning of rewards is preserved across stochasticity types within the same environment. Under this criterion:
>
>     - All Boxing stochasticity types and all BankHeist stochasticity types maintain reward consistency and are therefore directly comparable.
>
>     - For Breakout and Gopher, all stochasticity types except Type 2.1 satisfy this property and are eligible for cross-type comparison.
>
> - Why some stochasticity types break comparability:
>
>     Certain stochastic perturbations fundamentally alter the reward-generation process. For example, in Type 2.1 experiments for Breakout and Gopher, hits may be stochastically canceled. This artificially caps achievable returns even for an optimal agent, making these settings structurally incomparable to other stochastic types where optimal performance would yield higher reward.
>     Because of such reward distribution shifts, direct cross-type robustness comparisons would be misleading.
>
> Given these inconsistencies, our results section primarily reports relative algorithmic performance within each stochastic environment, which provides a more meaningful and fair basis for assessing robustness. We will incorporate a clear explanation of these constraints in the camera-ready version. Additionally, for all cases where reward semantics are preserved, we will include the full set of cross-type comparisons to address this concern more comprehensively.

---

### Meta-Review · Area_Chair_6yAF · 2025-12-30

**Summary:**

While reviewers agree that the paper is well motivated and that STORI could be a useful benchmark, the suggested decision is primarily driven by substantial concerns about scope, rigor, and empirical support. In particular, the paper frames itself as a benchmark for RL robustness but evaluates only two model-based agents, making the generality of the conclusions unclear. Additional issues around baseline comparability, mathematical underspecification, presentation quality, and the strength of some experimental claims further reduce confidence in the current version, leading reviewers to view the paper as below the acceptance threshold.

**Reviewer Concerns:**

The rebuttal successfully clarifies several points, including the rationale for environment selection, the intended baseline and comparability structure, and specific aspects of individual analyses. However, key concerns remain unresolved. Most notably, the evaluation is still restricted to a narrow class of model-based methods, and the paper’s claims go beyond what the experiments support. Furthermore, multiple issues related to mathematical precision, notation consistency, editing errors, and statistical robustness are acknowledged but remain unresolved in the current version. As a result, the rebuttal improves understanding but does not sufficiently address the reviewers' core weaknesses.

**Reviewer Scores:**

Reviewers who were initially mildly positive or borderline would likely maintain their scores or, at most, increase them slightly in response to the clarifications. Reviewers who expressed strong reservations about the limited methodological scope and overgeneralization of conclusions would likely remain negative, as the rebuttal does not introduce new evidence or analyses that directly resolve these concerns. Overall, the score distribution would likely remain weighted toward rejection.

---

### Decision · Program_Chairs · 2026-01-26

Reject